# Incidence, characteristics, and risk factors of new liver disorders 3.5 years post COVID-19 pandemic in the Montefiore Health System in Bronx

**Thomas Peng**[1☯], **Katie S. Duong**[1☯], **Justin Y. Lu**[1], **Kristina R. Chacko**[2], **Sonya Henry**[1], **Wei Hou**[1], **Kevin P. Fiori**[3], **Stephen H. Wang**[1,4], **Tim Q. Duong**[1]*

1 Department of Radiology, Albert Einstein College of Medicine and Montefiore Medical Center, Bronx, New York, United States of America, 2 Department of Medicine, Division of Hepatology, Albert Einstein College of Medicine and Montefiore Medical Center, Bronx, New York, United States of America, 3 Department of Pediatrics, Albert Einstein College of Medicine and Montefiore Medical Center, Bronx, New York, United States of America, 4 Department of Surgery, Beth Israel Deaconess Medical Center and Harvard Medical School, Boston, Massachusetts, United States of America

☯ These authors contributed equally to this work.
* tim.duong@einsteinmed.edu

## Abstract

### Purpose

To determine the incidence of newly diagnosed liver disorders (LD) up to 3.5-year post-acute COVID-19, and risk factors associated with new LD.

### Methods

We analyzed 54,699 COVID-19 patients and 1,409,547 non-COVID-19 controls from March-11-2020 to Jan-03-2023. New liver disorders included abnormal liver function tests, advanced liver failure, alcohol and non-alcohol related liver disorders, and cirrhosis. Comparisons were made with ambulatory non-COVID-19 patients and patients hospitalized for other lower respiratory tract infections (LRTI). Demographics, comorbidities, laboratory data, incomes, insurance status, and unmet social needs were tabulated. The primary outcome was new LD at least two weeks following COVID-19 positive test.

### Results

Incidence of new LD was not significantly different between COVID-19 and non-COVID-19 cohorts (incidence:1.99% vs 1.90% p>0.05, OR = 1.04[95%CI: 0.92,1.17], p = 0.53). COVID-19 patients with new LD were older, more likely to be Hispanic and had higher prevalence of diabetes, hypertension, chronic kidney disease, and obesity compared to patients without new LD. Hospitalized COVID-19 patients had no elevated risk of LD compared to hospitalized LRTI patients (2.90% vs 2.07%, p>0.05, OR = 1.29[0.98,1.69], p = 0.06). Among COVID-19 patients, those who developed LD had fewer patients with higher incomes (14.18% vs 18.35%, p<0.05) and more with lower incomes (21.72% vs 17.23%,

**Data Availability Statement:** The data underlying the results presented in the study are available via https://zenodo.org/records/10668558

**Funding:** The author(s) received no specific funding for this work.

**Competing interests:** The authors have declared that no competing interests exist.

p<0.01), more Medicare and less Medicaid insurance, and more patients with >3 unmet social needs (6.49% vs 2.98%, p<0.001) and fewer with no unmet social needs (76.19% vs 80.42%, p<0.001).

## Conclusions

Older age, Hispanic ethnicity, and obesity, but not COVID-19 status, posed increased risk for developing new LD. Lower socioeconomic status was associated with higher incidence of new LD.

## Introduction

Acute liver injury (ALI), a significant complication of coronavirus disease 2019 (COVID-19) [1, 2], has been associated with elevated risk of acute critical illness and mortality [3–6]. The SARS-CoV-2 coronavirus responsible for COVID-19 could directly infect liver cells [7]. Systemic hypoxia, sepsis, inflammation, and hepatotoxicity from antiviral, steroid and other medications used to treat acute SARS-CoV-2 infection could contribute to ALI indirectly [8–11]. Even if COVID-19 patients recovered fully from ALI, the insults from acute SARS-CoV-2 infection could increase long-term susceptibility to developing new liver disorders (LD) among COVID-19 survivors.

It is unknown if the systemic effects of SARS-CoV-2 infection result in long term new-onset LD. There could be long-lasting effects on the hepatic system after resolution of the acute infection, resulting in increased susceptibility to development of new-onset liver disorders among COVID-19 survivors. Other circumstances of the COVID-19 pandemic, including pandemic-associated weight gain, decreased access to health care providers and rising alcohol consumption [12, 13], increased the likelihood of new onset LD in all individuals. Given the large number of patients afflicted, it is important to identify if SARS-CoV-2 infection puts certain patients at increased risk of developing new-onset liver disorders. Identification of risk factors associated with development of new-onset LD would enable follow-up care for prevention or appropriate management of long-term liver dysfunction and complications.

In addition, socioeconomic inequities may play a role in disparities of COVID-19 related outcomes [14, 15]. Income inequality could result in unequal access and exacerbate the divide in health outcomes. Individuals of lower socioeconomic status face barriers in obtaining adequate healthcare, leading to delayed diagnoses, missed appointments, and worse outcomes [16]. Those lacking comprehensive coverage are also likely to be more affected by medical expenses. Moreover, social determinants of health, reported at the individual level as unmet social needs, such as housing quality, food insecurity, and limited access to transportation, could exert a substantial influence on health outcomes [17].

The goals of this study were to determine the incidence of newly diagnosed liver disorders up to 3.5 years post -acute COVID-19, identify risk factors associated with new-onset liver disorders among COVID-19 patients, and compare these measures against non-COVID controls. We also conducted analysis into socioeconomic factors relating to these outcomes, focusing on insurance status, median income, and unmet social needs.

## Methods

### Study setting

This retrospective study was approved by the Einstein-Montefiore Institutional Review Board (#2021–13658) with an exemption for informed consent and a HIPAA waiver and was performed in accordance with relevant guidelines and regulations. Data were obtained from March 11, 2020 to Jan 3, 2023. Authors no longer have access to information that could identify individual participants during or after data collection. The Montefiore Health System is one of the largest healthcare systems in New York City with multiple hospitals and outpatient clinics primarily located in the Bronx, the lower Hudson Valley, and Westchester County serving a large, low-income, and racially and ethnically diverse population that was heavily impacted by COVID-19 early in the pandemic and subsequent waves [18, 19].

### Data source

The data used were extracted as described previously [20, 21]. Deidentified EMR data were in the Observational Medical Outcomes Partnership (OMOP) Common Data Model (CDM) version 6. OMOP CDM stores the health data, which comes from many sources, into standard vocabulary concepts. This facilitates the systematic analysis of different observational databases, which includes data from the electronic medical record system, disease classification systems and administrative claims such as SNOWMED, ICD-10, LOINC, etc. Vocabulary concepts were then searched by ATLAS, a web-based tool that allows for navigation of patient-level, observational data in the CDM format developed by the Observational Health Data Sciences and Informatics (OHDSI) community, to build the cohort of patients. DB Browser for SQLite (version 3.12.0) was used to export and query data as SQLite database files. Data accuracy have been spotted checked and validated by chart review when needed as described previously [20–25].

### Cohorts

COVID-19 patients were defined by PCR test. All conditions used for inclusion/exclusion criteria and clinical outcomes were extracted using ICD-10 diagnosis codes entered into patient charts by billing providers. Liver disorders included ICD-10 definition of abnormal liver function tests (in the absence of a more specific diagnosis), advanced liver failure, alcohol and non-alcohol related liver disorders, and cirrhosis (S1 Table). Patients with these pre-existing LD, as well as any prior hepatitis B or C infection, were excluded, as the aim of the study was to identify patients with incident LD. Contemporary controls were patients without a COVID-19 test or those that tested negative over the same timeframe. Non-COVID-19 patients were matched to COVID-19 patients by visit month (1:1 match). No matching for additional factors such as sex or age was performed as these variables were already similar between groups. In addition, comparisons for the hospitalized COVID-19 cohort were made with patients hospitalized for lower respiratory tract infections (LRTI), including influenza, bronchiolitis, pneumonia, bronchitis, and other infections (S2 Table). Index dates for COVID-19 and LRTI admissions were similar, and no matching was needed. The same inclusion and exclusion criteria were applied.

### Clinical variables

Demographic data (e.g. age, sex, ethnicity, race), pre-existing comorbidities (congestive heart failure (CHF), chronic kidney disease (CKD), obesity, diabetes, chronic obstructive pulmonary disease (COPD), asthma), common laboratory tests (ALT, aspartate aminotransferase (AST), alkaline phosphatase (ALP), bilirubin, ferritin (FERR), lactate dehydrogenase (LDH), brain

natriuretic peptide (BNP), d-dimer (DDIM), and lymphocyte count ±7 days of the diagnosis date for hospitalized patients. Details regarding COVID-19 management including steroid use, invasive mechanical ventilation (IMV), and ICU admission status were also extracted. COVID-19 related hospitalizations were also tabulated.

Income quintiles, insurance status, and unmet social needs were obtained. Incomes, derived from zip code census data, were divided into quintiles. Insurance status was divided into private, CMO, Medicare, Medicaid, others, or uninsured. Unmet social needs consisted of voluntary responses to an eight-item questionnaire (S3 Table) that were implemented and collected in the EMR since April 2018. Only the responses to the questionnaires from the first visit were used. There were 171,922 unique patients in our overall cohort with responses to these questions. Scores divided into groups of 0, 1, 2–3 and >3 unmet social needs.

## Outcomes

The primary outcome was the incidence of new-onset LD more than two weeks following diagnosis of COVID-19.

## Statistical analysis

All statistical analyses were performed using Sklearn and Statsmodels Python packages and SAS. Group differences in disease frequency and percentages for categorical variables were tested using $\chi^2$ test. Demographics, comorbidities, and laboratory values, expressed as median (interquartile range, IQR), were compared between groups using analysis of variance (ANOVA) and post-hoc pairwise t-tests. Adjusted odds ratios (ORs) with 95% confidence intervals (CI) were estimated for each group in logistic regression adjusted for respective significant covariates. Two-tailed p-values <0.05 were considered statistically significant unless otherwise specified.

## Risk factors for incident LD

Logistic regression was used to identify the clinical variables associated with new-onset LD. Demographics (age, sex, race, and ethnicity), comorbidities, COVID-19 status, hospitalization status, and laboratory values at admission were used as input variables. ORs were computed for: i) COVID-19 patients only, ii) COVID-19 patients with non-COVID matched patients as reference, and iii) hospitalized-only COVID-19 patients with patients hospitalized for LRTI as reference.

## Results

### Patients with COVID-19 vs patients without COVID-19

Fig 1 describes the flowchart for patient selection for all patients. From March 2020 to January 2023, there were 54,699 patients who tested positive for COVID-19, of which 30,031 returned to our health system at least 2 weeks after COVID-19 positive date. Of those who returned, 2,032 patients had prior LD and were excluded. Of the remaining 27,993 patients with no prior LD, 557 (1.99%) developed new-onset LD. Many patients developed new LD soon after SARS-CoV-2 infection, with a median time to diagnosis of 6.0 months (IQR = 11 months; mean ± SD time-to-diagnosis 8.6 ±7.87 months) (S1 Fig).

There were 1,409,547 non-COVID-19 patients of which 604,021 patients returned to our health system. A contemporary non-COVID cohort was identified from these patients; after matching for visit date, 28,023 patients had no prior LD and never developed LD and 532

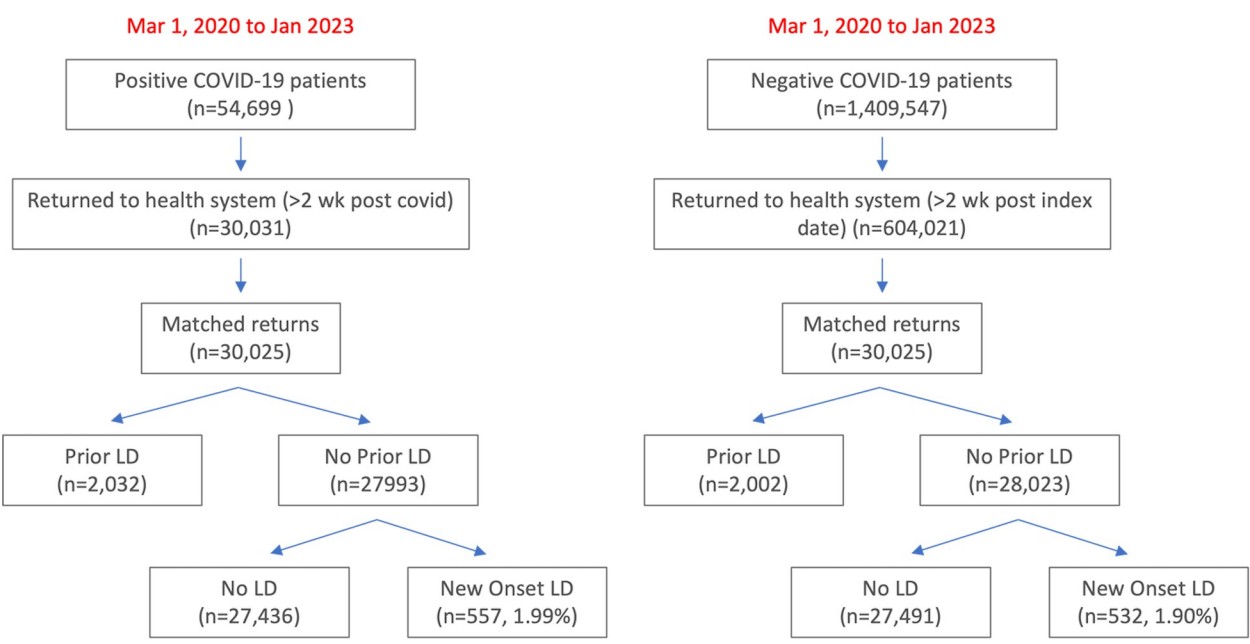

**Fig 1. Flowchart for patient selection of study patients from Mar 1, 2020 to Jan 2023.**

(1.90%) had newly diagnosed LD. There was no significant difference in incidence of new-onset LD between COVID and non-COVID cohorts (1.99% vs 1.90% p>0.05).

The distribution of incident liver disorders, stratified by COVID-19 status, is summarized in **S4 Table**. Abnormal liver function tests (ranging from 37.22–41.6% between cohorts) were the most common LD, followed by steatosis (28.99–35.9%), unspecified LD (20.23–23.68%), advanced liver failure (5.83–12.6%), and alcohol-related LD (2.63–5.8%). There were no significant differences in specific liver disorder distribution between COVID and non-COVID cohorts.

**Table 1** summarizes the demographics, comorbidities, insurance status, income quintiles for COVID-19 patients with new LD and no-LD compared to non-COVID controls. Among patients with COVID-19, patients with new onset LD tended to be older (53.49 vs 45.17 years old p<0.001) and more Hispanic (53.68 vs 42.45%, p<0.001), but less likely to be Black (28.31% vs 35.50%, p<0.01). COVID-19 survivors with incident LD were also more likely to have been hospitalized due to COVID-19 (44.63% vs 32.01%, p<0.001) and had higher prevalence of diabetes, hypertension, CKD, and obesity (p<0.05). Compared to patients without new LD, patients with incident LD were more likely to be on Medicare, less likely to be on Medicaid, with a higher % with lower (3rd quintile: $43,000-$59,999) incomes (21.72% vs 17.23%, p<0.01), lower % with higher (4th quintile: $60,000-$69,999) incomes (14.18% vs 18.35%, p<0.05), and higher % of >3 unmet social needs (6.49% vs 2.98%, p<0.001) and lower % with 0 unmet social need (76.19% vs 80.42%, p<0.001).

Among non-COVID-19 patients, patients with incident LD were older (54.81 vs 45.64 years old, p<0.001) and more likely to be male (41.92% vs 38.94%, p<0.05), white (p<0.01) or Asian (p<0.01), and less likely to be Black (p<0.01). Patients with incident LD had a higher prevalence of diabetes and hypertension (p<0.05), fewer patients on Medicaid, more patients on Medicare, more patients with highest income quintile, and fewer unmet social needs compared to non-COVID patients without incident LD.

**Table 1. Characteristics of patients with and without COVID-19 stratified by new LD.** $ indicates p<0.05, $ $ p<0.01, $ $ $ p<0.001 between LD and non-LD within group. * indicates p<0.05, ** p<0.01, *** p<0.001 between COVID and non-COVID incident LD groups. Note that the data for unmet social needs was only available in a subset of patients.

| | COVID+ | | COVID- | | p of COVID+ vs COVID- with LD |
|---|---|---|---|---|---|
| | LD (557, 1.99%) | No LD (27,436) | LD (532, 1.90%) | No LD (27,491) | |
| Age, mean±SD (yo) | 53.49 (18.72)$ $ $ | 45.17 (24.31) | 54.81 (18.65) $ $ $ | 45.64 (24.98) | |
| Male, n (%) | 220 (39.50%) | 11241 (40.97%) | 223 (41.92%)$ | 10704 (38.94%) | |
| Hispanic (%) | 299 (53.68%)$ $ $ | 11647 (42.45%) | 193 (36.28%) | 9495 (34.54%) | *** |
| Race | | | | | |
| White | 71 (13.22%) | 3587 (14.04%) | 140 (29.60%)$ $ | 5519 (23.27%) | *** |
| Black | 152 (28.31%)$ $ | 9069 (35.50%) | 118 (24.95%)$ $ | 7719 (32.55%) | |
| Asian | 14 (2.61%) | 914 (3.58%) | 25 (5.29%)$ $ | 724 (3.05%) | |
| Other | 300 (55.87%)$ $ $ | 11973 (46.87%) | 190 (40.17%) | 9754 (41.13%) | *** |
| % Hospitalized for covid | 262 (44.63%)$ $ $ | 8783 (32.01%) | N/A | N/A | |
| Comorbidities | | | | | |
| CHF | 24 (4.31%) | 1180 (4.30%) | 26 (4.89%) | 832 (3.03%) | |
| Diabetes | 128 (22.98%)$ $ $ | 4389 (16.00%) | 117 (21.99%)$ $ $ | 4178 (15.20%) | |
| Hypertension | 127 (22.80%)$ | 5059 (18.44%) | 127 (23.87%)$ | 5246 (19.08%) | |
| CKD | 53 (9.52%)$ $ | 1788 (6.52%) | 22 (4.13%) | 1318 (4.79%) | *** |
| Obesity | 102 (18.31%)$ $ $ | 3608 (13.15%) | 89 (16.73%)$ | 3891 (14.15%) | |
| Medical Insurance n (%) | | | | | |
| Private | 144 (28.02%) | 7395 (28.79%) | 152 (30.58%) | 7447 (29.29%) | |
| Medicaid | 168 (32.68%)$ $ $ | 11016 (42.89%) | 147 (29.58%)$ $ $ | 10154 (39.93%) | |
| Medicare | 176 (34.24%)$ $ $ | 6249 (24.33%) | 163 (32.80%)$ $ $ | 6571 (25.84%) | |
| Uninsured | 0 (0.00%) | 5 (0.02%) | 0 (0.00%) | 1 (0.00%) | |
| CMO | 25 (4.86%) | 893 (3.48%) | 35 (7.04%)$ | 1227 (4.83%) | |
| Other | 1 (0.19%) | 127 (0.49%) | 0 (0.00%) | 29 (0.11%) | |
| Income quintiles, n (%) | | | | | |
| < = 34,999 | 80 (14.36%) | 3721 (13.59%) | 68 (12.81%) | 4080 (14.88%) | |
| 35,000–42,999 | 155 (27.83%) | 7572 (27.66%) | 103 (19.40%) | 6148 (22.43%) | ** |
| 43,000–59,999 | 121 (21.72%)$ $ | 4718 (17.23%) | 102 (19.21%)$ | 4364 (15.92%) | |
| 60,000–69,999 | 79 (14.18%)$ | 5025 (18.35%) | 52 (9.79%)$ | 3633 (13.25%) | * |
| > = 70,000 | 122 (21.90%) | 6344 (23.17%) | 206 (38.79%)$ | 9189 (33.52%) | *** |
| Unmet social needs n (%) | | | | | |
| 0 | 176 (76.19%)$ $ $ | 5935 (80.42%) | 140 (78.65%)$ $ | 5867 (82.60%) | |
| 1 | 20 (8.66%) | 696 (9.43%) | 20 (11.24%)$ | 640 (9.01%) | |
| 2–3 | 20 (8.66%)$ $ | 529 (7.17%) | 12 (6.74%) | 415 (5.84%) | |
| >3 | 15 (6.49%)$ $ $ | 220 (2.98%) | 6 (3.37%) | 181 (2.55%) | |

Between COVID-19 and non-COVID-19 patients with incident LD, the COVID-19 cohort had more Hispanic (53.68% vs 36.28%, p<0.001) and fewer White (p<0.001) patients, and a higher prevalence of pre-existing CKD (p<0.001). Patients with COVID-19 were more likely to fall into the lower income quintiles, but there was no difference in the unmet social needs.

Odds ratios: Multiple regression analysis showed that among COVID-19 patients, age, Hispanic ethnicity, hospitalization status, obesity and Medicaid were significant factors associated with developing LD (**Table 2A**). When all study patients were included in the regression analysis (**Table 2B**), age, Hispanic ethnicity, diabetes, and obesity were associated with greater risk of developing LD. COVID-19 status, however, was not associated with developing LD (OR = 1.04 [95%CI: 0.92,1.17], p = 0.53).

**Table 2. Odds Ratios of developing LD using (A) patients with COVID-19 only and (B) all patients with and without COVID-19 data.** Input data include all those shown in Table 1 except unmet social needs, insurance status, and median incomes. For income, 5ᵗʰ quintile (highest) was used as reference. For insurance, private was used as reference.

**(A) Patients with COVID-19**

| | OR | 2.50% | 97.50% | p-value |
|---|---|---|---|---|
| Age | 1.01 | 1.01 | 1.02 | <0.001 |
| Male | 0.96 | 0.80 | 1.14 | 0.60 |
| Hispanic | 1.42 | 1.09 | 1.84 | 0.008 |
| Race—Black | 1.05 | 0.79 | 1.38 | 0.74 |
| Race—Asian | 1.05 | 0.59 | 1.87 | 0.85 |
| Race—Other | 1.31 | 0.98 | 1.75 | 0.06 |
| Hospitalization | 1.55 | 1.28 | 1.87 | <0.001 |
| Congestive Heart Failure | 0.66 | 0.43 | 1.02 | 0.06 |
| Diabetes | 1.14 | 0.91 | 1.42 | 0.25 |
| Hypertension | 0.89 | 0.71 | 1.12 | 0.30 |
| Chronic Kidney Disease | 1.08 | 0.79 | 1.49 | 0.61 |
| Obesity | 1.43 | 1.14 | 1.80 | 0.002 |
| Income 1st Quintile | 1.03 | 0.76 | 1.39 | 0.85 |
| Income 2nd Quintile | 0.94 | 0.73 | 1.21 | 0.62 |
| Income 3rd Quintile | 1.17 | 0.90 | 1.52 | 0.24 |
| Income 4th Quintile | 0.79 | 0.59 | 1.06 | 0.11 |
| Insurance—Medicaid | 0.77 | 0.61 | 0.96 | 0.02 |
| Insurance—Medicare | 0.82 | 0.64 | 1.05 | 0.12 |
| Insurance—Other | 1.22 | 0.80 | 1.86 | 0.34 |

**(B) All patients**

| | OR | 2.50% | 97.50% | p-value |
|---|---|---|---|---|
| Age | 1.02 | 1.01 | 1.02 | <0.001 |
| Male | 1.08 | 0.96 | 1.22 | 0.20 |
| Hispanic | 1.28 | 1.07 | 1.53 | 0.007 |
| Race—Black | 0.87 | 0.73 | 1.04 | 0.12 |
| Race—Asian | 1.40 | 0.99 | 1.97 | 0.056 |
| Race—Other | 1.08 | 0.90 | 1.30 | 0.42 |
| Congestive Heart Failure | 0.90 | 0.67 | 1.21 | 0.48 |
| Diabetes | 1.20 | 1.02 | 1.41 | 0.02 |
| Hypertension | 0.91 | 0.78 | 1.07 | 0.26 |
| Chronic Kidney Disease | 0.84 | 0.65 | 1.08 | 0.17 |
| Obesity | 1.34 | 1.14 | 1.58 | 0.001 |
| COVID-19 Status | 1.04 | 0.92 | 1.17 | 0.53 |

## Hospitalized patients with COVID-19 vs LRTI

It is possible patients hospitalized for COVID-19 might have higher incidence of new LD and we thus compared patients hospitalized for COVID-19 with those hospitalized for LRTI. Fig 2 describes the flowchart for hospitalized patient selection as a sub-analysis. From March 2020 to January 2023, there were 17,015 hospitalized patients with COVID-19, of which 9,887 returned to our health system at least 2 weeks later. Of those who returned, 841 patients were excluded due to prior LD. Of the remaining 9,046 patients, 262 (2.90%) patients developed new-onset LD.

Comparison was made with a hospitalized LRTI+ cohort over the same time period to evaluate how the effects of COVID-19 were compared with other respiratory infections. There

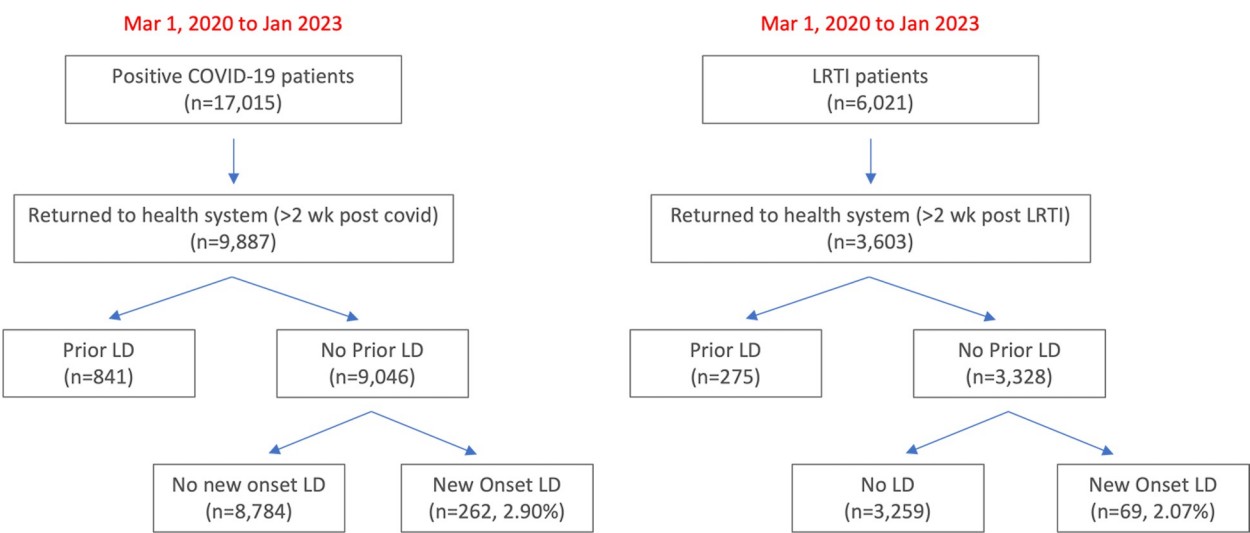

**Fig 2. Flowchart for patient selection of patients hospitalized for COVID-19 or LRTI from Mar 1, 2020 to Jan 2023.**

were 6,021 patients hospitalized for LRTI, of which 3,603 returned to our health system at least 2 weeks post-infection. Of the those who returned, 275 patients were excluded for prior LD. Of the remaining 3,328 patients, 69 (2.07%) had new onset LD. The incidence of new-onset LD was significantly higher among patients hospitalized for COVID-19 compared to LRTI (2.90% vs 2.07%, p<0.05).

Table 3 summarizes the demographics, comorbidities, insurance status, income quintiles, laboratory data at admission, for patients hospitalized for COVID-19 or LRTI and stratified by incident LD status. Among patients hospitalized with COVID-19, those with new LD tended to be Hispanic (55.34% vs 41.91%, p<0.001), with no differences in comorbidities, but with higher ALT, AST, CRP, ferritin compared to the no-LD group (p<0.05). Among hospitalized patients with LRTIs, the incident LD cohort also had more Hispanic patients (53.62% vs 37.62%, p<0.01) and higher ALT and AST lab values compared to no-LD group. Between patients with incident LD hospitalized for COVID-19 vs LRTI, there were fewer White patients and higher prevalence of diabetes in the COVID-19 cohort (p<0.05). Ferritin, LDH, BNP and WBC values at admission were significantly different between groups (p<0.05).

Table 4 presents multiple regression analysis of patients hospitalized with COVID-19 or LRTI. Hispanic ethnicity (OR 1.46, 95% CI [1.02, 2.07], p = 0.03) and Other races (OR 1.61 [1.08, 2.41], p = 0.02) were associated with incident LD, while other demographics, comorbidities, and COVID-19 status (OR = 1.29[0.98,1.69], p = 0.06) were not significantly associated with development of incident LD.

## Discussion

This study investigated the incidence and risk factors of newly diagnosed liver disorders up to 3.5 years following SARS-CoV-2 infection compared to matched non-COVID patients. The major findings are: i) the incidence of new LD across different cohorts ranged from 1.90 to 1.99% and was not different between COVID-19 and non-COVID cohorts, ii) among patients with COVID-19, patients with incident LD were older, more likely to be Hispanic, hospitalized for COVID-19, with a higher prevalence of co-morbid conditions including diabetes, hypertension, CKD, and obesity, and have higher proportion of low income quintiles and unmet social needs, compared to patients without incident LD, iii) patients hospitalized for COVID-

**Table 3. Characteristics of patients hospitalized for COVID-19 or LRTI with and without new-onset LD.** $ indicates p<0.05, $ $ p<0.01, $ $ $ p<0.001 between LD and non-LD. * indicates p<0.05, ** p<0.01, *** p<0.001 between covid+ and LRTI. LRTI: lower-tract respiratory infection: ICU: intensive care unit; IMV: invasive mechanical ventilation.

| | COVID+ | | LRTI | | P of COVID+ vs LRTI with LD |
|---|---|---|---|---|---|
| | LD (262, 2.91%) | No LD (8,784) | LD (n = 69, 2.07%) | No LD (n = 3259) | |
| Age, mean (SD) | 58.06 (18.69)$ | 59.09 (21.83) | 58.98 (21.40) | 53.57 (31.54) | |
| Male, n (%) | 127 (48.47%) | 3964 (45.13%) | 29 (42.03%) | 1624 (49.83%) | |
| Hispanic (%) | 145 (55.34%)$ $ $ | 3681 (41.91%) | 37 (53.62%)$ $ | 1226 (37.62%) | |
| Race | | | | | |
| White | 26 (9.92%) | 1200 (13.66%) | 15 (21.74%) | 809 (26.10%) | * |
| Black | 75 (28.63%)$ | 3175 (36.15%) | 19 (27.54%) | 988 (31.87%) | |
| Asian | 7 (2.67%) | 281 (3.2%) | 0 (0%) | 75 (2.42%) | |
| Other | 150 (57.25%)$ $ $ | 3784 (43.08%) | 35 (50.72%)$ | 1228 (39.61%) | |
| Comorbidities | | | | | |
| CHF | 18 (6.87%) | 776 (8.83%) | 3 (4.35%) | 231 (7.09%) | |
| Diabetes | 69 (26.34%) | 2138 (24.34%) | 9 (13.04%) | 488 (14.97%) | * |
| Hypertension | 62 (23.66%) | 2566 (29.21%) | 21 (30.43%) | 808 (24.79%) | |
| CKD | 30 (11.45%) | 1125 (12.81%) | 5 (7.25%) | 307 (9.42%) | |
| Obesity | 44 (16.79%) | 1176 (13.39%) | 5 (7.25%) | 218 (6.69%) | |
| Steroid use | 114 (43.51%)$ $ | 3080 (35.06%) | 22 (31.88%) | 1047 (32.13%) | |
| Critical Illness (ICU or IMV) | 27 (10.31%) | 787 (8.96%) | 5 (7.25%) | 285 (8.75%) | |
| Lab values @ admission, Median (IQR) | | | | | |
| ALT | 32.0 (20.0, 58.0)$ $ $ | 22.0 (15.0, 37.0) | 27.00 (20.25, 60.75)$ $ $ | 18.00 (12.00, 30.00) | |
| AST | 39.0 (25.0, 66.25)$ $ $ | 28.0 (20.0, 43.0) | 29.50 (20.5, 45.75)$ $ | 23.00 (20.00, 34.00) | |
| CRP | 5.2 (2.0, 11.2)$ | 4.4 (1.4, 9.9) | 5.45 (1.50–10.50) | 5.40 (1.30, 15.50) | |
| Ferritin | 566.4 (228.1, 1316.45)$ $ | 421.7 (174.1, 967.6) | 116.10 (38.48, 249.83) | 324.00 (121.00, 817.00) | * |
| LDH | 334.0 (245.0, 445.0)$ $ $ | 292.0 (220.0, 403.0) | 245.00 (207.50, 260.75) | 262.00 (202.25, 361.50) | ** |
| BNP | 60.0 (19.0, 334.25)$ | 99.0 (31.0, 427.0) | 192.50 (70.25, 429.00) | 153.00 (46.00, 575.50) | * |
| Cr | 0.85 (0.71, 1.14)$ | 0.9 (0.71, 1.3) | 0.89 (0.64, 1.08) | 0.80 (0.60, 1.16) | |
| D-dimer | 0.98 (0.6, 1.93) | 1.07 (0.57, 2.19) | 2.05 (0.75, 3.25) | 2.09 (0.85, 14.42) | |
| TnT | 0.01 (0.01, 0.01)$ $ | 0.01 (0.01, 0.03) | 0.01 (0.01, 0.02) | 0.01 (0.01, 0.02) | |
| WBC | 6.6 (4.9, 9.0) | 6.8 (5.0, 9.5) | 7.50 (5.73, 11.20) | 8.70 (6.20, 12.00) | ** |
| LYMPH | 17.5 (11.0, 27.0) | 19.0 (12.0, 28.0) | 15.70 (10.50, 26.00) | 17.30 (10.30, 27.00) | |

19 with incident LD were less likely to be white but had greater prevalence of diabetes, higher levels of ferritin, LDH, BNP and lower WBC compared to patients hospitalized for LRTI and developed new LD, iv) multiple regression analysis showed that COVID-19 status was not significantly associated with development of new LD when compared to both contemporary non-COVID controls and contemporary patients hospitalized for LRTIs. Thus, we concluded that COVID-19 does not confer increased risk for developing new LD.

Acute liver injury has been reported to be associated with higher mortality [3–6, 26, 27]. The overall incidence rate of ALI in COVID-19 is relatively low [26, 27] compared to acute cardiac injury and acute kidney injury [19, 21, 22]. Previous studies found that most patients with COVID-19 and ALI had normalized AST and ALT values at discharge and 2.5 months after hospitalization [26], whereas lab abnormalities in acute cardiac injury and acute kidney injury took longer to recover at a cohort level [19, 26]. Independent of whether or not patients with COVID-19 recovered from ALI during the acute infection, the hepatic insults associated with COVID-19 could increase long-term susceptibility to developing new LD among survivors.

**Table 4. Odds ratios of LD for (A) Both hospitalized COVID-19 and LRTI patients and (B) All hospitalized COVID-19 patients.**

| (A) Both hospitalized COVID-19 and LRTI patients (n = 12,374) | | | | |
|---|---|---|---|---|
| | OR | 2.50% | 97.50% | p-value |
| Age | 1.00 | 1.00 | 1.01 | 0.09 |
| Male | 1.08 | 0.87 | 1.35 | 0.49 |
| Hispanic | 1.46 | 1.02 | 2.07 | 0.03 |
| Black | 1.34 | 0.93 | 1.94 | 0.12 |
| Asian | 1.18 | 0.53 | 2.67 | 0.68 |
| Other | 1.61 | 1.08 | 2.41 | 0.02 |
| Congestive Heart Failure | 0.74 | 0.47 | 1.17 | 0.19 |
| Diabetes | 1.09 | 0.82 | 1.45 | 0.56 |
| Hypertension | 0.82 | 0.62 | 1.08 | 0.16 |
| Chronic Kidney Disease | 0.88 | 0.60 | 1.29 | 0.50 |
| Obesity | 1.33 | 0.97 | 1.84 | 0.08 |
| COVID-19 Status | 1.29 | 0.98 | 1.69 | 0.06 |
| (B) All hospitalized COVID-19 patients (n = 9,046) | | | | |
| | OR | 2.50% | 97.50% | p-value |
| Age | 1.00 | 1.00 | 1.01 | 0.68 |
| Male | 1.19 | 0.92 | 1.52 | 0.18 |
| Hispanic | 1.36 | 0.91 | 2.01 | 0.13 |
| Black | 1.32 | 0.85 | 2.05 | 0.21 |
| Asian | 1.39 | 0.60 | 3.23 | 0.44 |
| Other | 1.69 | 1.06 | 2.70 | 0.02 |
| Congestive Heart Failure | 0.80 | 0.49 | 1.33 | 0.39 |
| Diabetes | 1.19 | 0.88 | 1.62 | 0.26 |
| Hypertension | 0.72 | 0.52 | 0.99 | 0.04 |
| Chronic Kidney Disease | 0.94 | 0.62 | 1.43 | 0.77 |
| Obesity | 1.36 | 0.97 | 1.93 | 0.07 |

To our knowledge this is the first cohort study investigating new-onset LD among patients with COVID-19. Abnormal liver function tests (in the absence of a more specific diagnosis) and hepatic steatosis constituted the majority of new-onset LD, with no significant differences in the type of new LD between the COVID-19 group and non-COVID-19 group. We found that patients with COVID-19 who developed new LD tended to be older, more obese, and of Hispanic ethnicity, all of which overlap with known risk factors for both alcohol-related and metabolic-associated fatty liver disease [28–31]. Other factors, such as the prevalence of metabolic syndrome, type 2 diabetes, access to health care, and socioeconomic status, also play a role in the development of liver disease [32]. Our analysis showed no significant association of COVID status with development of new LD in all patients with COVID-19 compared to those without COVID-19, as well as among patients hospitalized with COVID-19 or LRTIs. These findings indicate that while certain risk factors may be associated with the development of LD in patients with COVID-19, SARS-CoV-2 infection alone is unlikely to be a significant driver of this outcome.

Among the hospitalized patients, the COVID-19 cohort had higher ferritin and LDH, and lower BNP and WBC at admission compared to the hospitalized LRTI cohorts. These differences in laboratory parameters could be indicative of unique disease mechanisms or inflammatory responses in COVID-19 [9–11, 33–38]. There were generally higher levels of AST and ALT at admission in patients hospitalized for COVID-19 compared to those hospitalized for

LRTI. The median values for AST and ALT in incident LD cohorts were higher than no-new LD cohorts but overall within normal levels, suggesting that any lasting injury to hepatocytes may be more insidious and not immediately evident during the acute illness.

## Social determinants of health

The COVID-19 pandemic has highlighted the negative impact of socioeconomic determinants of health on the spread and outcomes of COVID-19 [14, 15]. Socioeconomic factors such as income, insurance status, and unmet social needs could play a pivotal role in shaping the vulnerability of individuals and communities. Lower-income populations often faced challenges in adhering to preventive measures due to overcrowded living conditions, limited access to quality healthcare, and greater reliance on essential jobs that increase exposure risk for COVID-19. Additionally, individuals in marginalized communities may experience higher levels of chronic stress, which can weaken immune system and increase vulnerability to diseases [39].

We analyzed the interplay between three socioeconomic factors (income levels, insurance statuses, and unmet social needs) and the incidence of new-onset LD in the context of COVID-19. While not uniformly consistent across all income quintiles, there was a general trend of fewer patients in higher income brackets for patients with new-onset LD.

Among patients with COVID-19, those who experienced the development of new LD exhibited a higher prevalence of underinsured statuses, predominantly characterized by reliance on Medicare. It is important to note that patients on Medicare were generally older and thus expected to be more vulnerable to developing new LD. Patients with COVID-19 had a lower percentage of individuals covered by private insurance and a higher proportion reliant on Medicare and Medicaid, though some of the difference may be due to the greater mean age of the COVID-19 cohort. This dichotomy was further underscored by their income distribution, which skewed towards lower income quintiles.

In a subset of patients who had documented surveys on unmet socioeconomic needs, strikingly, a higher percentage of individuals grappling with more unmet social needs was evident in the COVID-19 cohort when contrasted with patients who did not develop LD. Similar patterns were found in the non-COVID cohort, with a notable exception: no significant difference in patients with a greater number of unmet social needs compared to non-LD counterparts. These findings suggest a significant influence of socioeconomic determinants of health on the emergence of new LD among survivors of COVID-19, consistent with the pattern of COVID-19 exacerbating health disparities. Collectively, these findings underscore the negative impact of socioeconomic determinants of health on the long-term outcomes of COVID-19.

## Limitations

There are several limitations in our analysis. Our study is limited by the lack of data on home testing or SARS-CoV-2 infection which was not documented by a positive PCR test which may result in undercounting of patients with COVID-19. This limits our study to showing significance in patients who had more severe COVID-19 symptoms which elicited PCR testing and clinical evaluation.

Furthermore, our findings are limited to patients who returned to our health system. It is possible that patients who returned had more severe disease, prompting increased healthcare utilization. However, the patient data obtained via EMR included those who returned for any medical reason to a large health system serving predominant the Bronx and its environs, including routine medical visits. It is also possible that some patients had previously undiagnosed LD prior to COVID-19 or LRTI which could result in misclassification.

The incidence of LD across the pandemic might be affected by other factors including vaccination rate, specific strain of SARS-CoV-2, testing rate, population profile, and disease severity. Vaccine status was not reliably recorded if patients received vaccine outside our healthcare system. Vaccines and boosters were also administered in multiple stages based on age, and multiple doses, and types (some requiring one shot and others two shots) in the population. Thus, vaccination status is difficult to analyze with respect to outcomes. COVID-19 testing rate and the profile of the patients (i.e., more older patients were affected earlier in the pandemic) could also affect the temporal incidence of new-onset LD. D-dimer, CRP and other biomarkers during acute COVID-19 could also affect incidence of LD [40]. The effects of these confounders on outcomes are complex, difficult to assess, and not readily discernable from one another. We followed patients for 3.5 years after diagnosis, but a longer follow-up study would be needed. Although data on unmet social needs are unique, not all patients voluntarily responded causing a sampling bias. Data was also limited regarding the causative organisms in the LRTI cohort [41]. The vast majority of LRTI patients did not have a specific infectious agent specified, and thus any differences in outcomes between bacterial, viral, and fungal etiologies were unable to be investigated. As with any retrospective study, there could be other unintended patient selection bias and latent confounders.

## Conclusions

This study provides important insights into the relationship between COVID-19 and new-onset liver disorders. In these data, COVID-19 status was not associated with increased risk for developing new liver disorders. Among patients with COVID-19, older age, Hispanic ethnicity, and obesity were associated with increased risk for developing new LD. New LD in survivors of COVID-19 was associated with lower socioeconomic status, underscoring the negative impact of socioeconomic determinants of health on the long-term outcomes of COVID-19. Understanding the impact of COVID-19 on liver health is crucial for optimizing patient care and developing targeted interventions to reduce the burden of liver disorders, especially in socioeconomically disadvantaged populations.

## Supporting information

**S1 Table. LD and OMOP concept ids.**
(DOCX)

**S2 Table. LRTI and OMOP concept ids.**
(DOCX)

**S3 Table. Social determinant of health questionnaires.** Respondents answered yes or no to each question.
(DOCX)

**S4 Table. Distribution of incident LD.** These percentages are not mutually exclusive and do add up more than 100% (i.e., patients could have multiple disorders). There were no statistical significances between groups.
(DOCX)

**S1 Fig. A histogram of time to diagnosis of incident liver disorders from index date for COVID-19 survivors.**
(PNG)

## Acknowledgments

Consent to participate

Consent was waived as this retrospective study was approved by the Einstein-Montefiore Institutional Review Board (#2021–13658) with an exemption for informed consent and a HIPAA waiver.

## Author Contributions

**Conceptualization:** Justin Y. Lu, Stephen H. Wang, Tim Q. Duong.

**Data curation:** Thomas Peng, Katie S. Duong, Justin Y. Lu, Sonya Henry, Kevin P. Fiori.

**Formal analysis:** Thomas Peng, Katie S. Duong, Wei Hou.

**Investigation:** Thomas Peng, Katie S. Duong, Kristina R. Chacko, Stephen H. Wang.

**Methodology:** Justin Y. Lu, Sonya Henry, Wei Hou, Stephen H. Wang.

**Project administration:** Tim Q. Duong.

**Resources:** Kevin P. Fiori.

**Supervision:** Tim Q. Duong.

**Validation:** Stephen H. Wang.

**Writing – original draft:** Thomas Peng, Katie S. Duong.

**Writing – review & editing:** Kristina R. Chacko, Stephen H. Wang, Tim Q. Duong.

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
