## [Decision Letter · Decision Letter 0]

5 Mar 2024

PONE-D-23-43546Incidence, characteristics, and risk factors of new liver disorders 3.5 years post COVID-19 pandemic in the Montefiore Health System in Bronx

PLOS ONE

Dear Dr. Duong,

Thank you for submitting your manuscript to PLOS ONE. After careful consideration, we feel that it has merit but does not fully meet PLOS ONE’s publication criteria as it currently stands. Therefore, we invite you to submit a revised version of the manuscript that addresses the points raised during the review process.

Editor's comments:

1. Typo "Acute livery injury" should be "Acute liver injury"

2. Please clarify P value is one-tail or two-tail.

3. What are diagnostic tools that are used to rule out viral hepatitis ? this should be included in the manuscript.

4. What are the etiology of the new-onset LD? this should be included in the manuscript.

5. How to detect LRTI should be included in the manuscript

6. The authors should expand the discussion based on the statement "The incidence of LD across the pandemic might be affected by other factors including vaccination rate, specific strain of SARS-CoV-2, testing rate, population profile, and disease severity." The authors should discuss the overlapped risk factors between LD and severe COVID-19 as well as the significance of D-dimer, CRP in COVID. More references should be cited, with the following as an example (citing is optional):

Clinical significance of measuring serum cytokine levels as inflammatory biomarkers in adult and pediatric COVID-19 cases: A review. Cytokine. 2021 Jun;142:155478. doi: 10.1016/j.cyto.2021.155478. Epub 2021 Feb 23. PMID: 33667962; PMCID: PMC7901304.

7. As for LRTI, a limited organisms were looked at in Table. The authors should discuss that bacterial or fungal comorbidities may be contributing factors to the difference of LD vs non-LD groups. Unbalanced distribution rate of respiratory pathogens/organisms in LD group and Non-LD group may lead to the difference between the both groups, for example, opportunistic fungi (e.g., Pneumocystis jirovecii). For example, Pneumocystis jirovecii tend to colonize in the lungs and lower respiratory tract in LRTI patients. This study under review may not look at the impact of Pneumocystis jirovecii on LD group vs Non-LD group using molecular diagnostic test for Pneumocystis jirovecii, respectively. The above mentioned discussion should be made in Discussion and more references are recommended to cite, with the following as example (citing suggestion is optional).

Development and Evaluation of a Fully Automated Molecular Assay Targeting the Mitochondrial Small Subunit rRNA Gene for the Detection of Pneumocystis jirovecii in Bronchoalveolar Lavage Fluid Specimens. J Mol Diagn. 2020 Dec;22(12):1482-1493. doi: 10.1016/j.jmoldx.2020.10.003. Epub 2020 Oct 15. PMID: 33069878.

We look forward to receiving your revised manuscript.

Kind regards,

Benjamin M. Liu, MBBS, PhD, D(ABMM), MB(ASCP)

Academic Editor

PLOS ONE

Journal Requirements:

3. We note that you have referenced (Ho SL, Lu JQ, Buczek A, et al) which has currently not yet been accepted for publication. Please remove this from your References and amend this to state in the body of your manuscript: (Ho SL, Lu JQ, Buczek A, et al. [Submitted]”) as detailed online in our guide for authors

4. We notice that your supplementary figures and tables are included in the manuscript file. Please remove them and upload them with the file type 'Supporting Information'. Please ensure that each Supporting Information file has a legend listed in the manuscript after the references list.

Reviewers' comments:

Reviewer's Responses to Questions

**Comments to the Author**

1. Is the manuscript technically sound, and do the data support the conclusions?

Reviewer #1: No

2. Has the statistical analysis been performed appropriately and rigorously? 

Reviewer #1: No

3. Have the authors made all data underlying the findings in their manuscript fully available?

Reviewer #1: No

4. Is the manuscript presented in an intelligible fashion and written in standard English?

Reviewer #1: Yes

5. Review Comments to the Author

Reviewer #1: From the perspective of clinical real-world observation, this study conducted a detailed comparative analysis of population characteristics, basic diseases, laboratory examination indicators, a retrospective analysis of new liver diseases, and also an analysis of socio-economic factors, between COVID-19 and non-COVID-19 patients, as well as between COVID-19 patients and lower respiratory tract infected patients.

These analyses provide targeted data support for the prevention and management of newly diagnosed liver disease patients in the post pandemic era, and have very important practical significance.

It is almost a pleasant thing to read the text section of this article separately, except for the results section of abstract, which has a spelling error of "laower" instead of "lower".

However, the numerical values described in the results section of this article do not match the values in the chart or the abstract section, to the extent that they cannot be described in language. Listing them would be a huge workload and meaningless.

The numerical values of the main text, charts, and abstract, like three different studies, overshadow the topic which has great practical significance originally.

Considering the large number of patients enrolled in this study and the long observation time, which may lead to a large number of institutions and personnel involved in this study, there may be difficulties in unified coordination in reality. Therefore, the authenticity of the data will not be discussed temporarily.

Researchers must select a representative, read and review through the text, charts, and abstract, and confirm the consistency of the numerical descriptions of the three.

1. This article lacks the table or graph of multiple regression analysis of COVID-19 and LRTI inpatients, which is similar to Table 2 showing the regression analysis of COVID-19 and non-COVID-19 patients.

2. How can the median diagnostic time determined to be 6 months in the presentation of Supplementary Figure 1?

6. PLOS authors have the option to publish the peer review history of their article (what does this mean?). If published, this will include your full peer review and any attached files.

Reviewer #1: No

---

## [Author Response · Author response to Decision Letter 0]

1 Apr 2024

PONE-D-23-43546

Incidence, characteristics, and risk factors of new liver disorders 3.5 years post COVID-19 pandemic in the Montefiore Health System in Bronx

Dear Editor and reviewers, 

 We thank you for your careful review of our manuscript. Please find the point-by-point responses and the revised manuscript. 

Sincerely, Tim Duong. 

Editor's comments:

1. Typo "Acute livery injury" should be "Acute liver injury"

Thank you for your comment. We have performed extensive revisions to correct spelling and grammatical errors such as this one throughout the manuscript.

2. Please clarify P value is one-tail or two-tail.

Thank you for your comment. We have revised the Statistical Analysis section in the Methods to state “Two-tailed p-values…”.

3. What are diagnostic tools that are used to rule out viral hepatitis ? this should be included in the manuscript.

Thank you for your comment. All diagnoses were based on ICD-10 codes entered into patient charts for the conditions listed in Supplemental Tables 1 and 2. Thus, any conditions used for inclusion or exclusion criteria, or outcomes, were determined based on the treating providers’ billed diagnoses and not specific diagnostic studies or lab values. We have also clarified this further in the Methods section.

4. What are the etiology of the new-onset LD? this should be included in the manuscript.

Thank you for your comment. We have added Supplementary table 4 to show the potential different LD of etiology for the 4 groups, added a paragraph about the distribution in the Results, and a discussion to reflect your comments.

5. How to detect LRTI should be included in the manuscript

All diagnoses were based on ICD-10 codes entered into patient charts for the conditions listed in Supplemental Tables 1 and 2. Thus, any conditions meeting inclusion or exclusion criteria, or outcomes, were determined based on the treating providers’ billed diagnoses and not specific diagnostic studies or lab values. We have also clarified this further in the Methods section.

6. The authors should expand the discussion based on the statement "The incidence of LD across the pandemic might be affected by other factors including vaccination rate, specific strain of SARS-CoV-2, testing rate, population profile, and disease severity." The authors should discuss the overlapped risk factors between LD and severe COVID-19 as well as the significance of D-dimer, CRP in COVID. More references should be cited, with the following as an example (citing is optional):

Clinical significance of measuring serum cytokine levels as inflammatory biomarkers in adult and pediatric COVID-19 cases: A review. Cytokine. 2021 Jun;142:155478. doi: 10.1016/j.cyto.2021.155478. Epub 2021 Feb 23. PMID: 33667962; PMCID: PMC7901304.

Thank you for the reference. It is cited. The following is added

The incidence of LD across the pandemic might be affected by other factors including vaccination rate, specific strain of SARS-CoV-2, testing rate, population profile, and COVID-19 disease severity. In particular, D-dimer, CRP and other biomarkers during acute COVID-19 could also affect incidence of LD [1].

7. As for LRTI, a limited organisms were looked at in Table. The authors should discuss that bacterial or fungal comorbidities may be contributing factors to the difference of LD vs non-LD groups. Unbalanced distribution rate of respiratory pathogens/organisms in LD group and Non-LD group may lead to the difference between the both groups, for example, opportunistic fungi (e.g., Pneumocystis jirovecii). For example, Pneumocystis jirovecii tend to colonize in the lungs and lower respiratory tract in LRTI patients. This study under review may not look at the impact of Pneumocystis jirovecii on LD group vs Non-LD group using molecular diagnostic test for Pneumocystis jirovecii, respectively. The above mentioned discussion should be made in Discussion and more references are recommended to cite, with the following as example (citing suggestion is optional).

Development and Evaluation of a Fully Automated Molecular Assay Targeting the Mitochondrial Small Subunit rRNA Gene for the Detection of Pneumocystis jirovecii in Bronchoalveolar Lavage Fluid Specimens. J Mol Diagn. 2020 Dec;22(12):1482-1493. doi: 10.1016/j.jmoldx.2020.10.003. Epub 2020 Oct 15. PMID: 33069878.

Thank you for the reference; it is cited. We examined the distribution of patients stratified by LRTI and have included this in Supplemental Table 2. The vast majority of patients were hospitalized for pneumonia without a specific organism identified. Of the patients with an identified organism, the distribution favored viral etiologies including influenza, RSV, and rhinovirus over bacterial causes. Due to the lack of further clarity on LRTI etiologies from the diagnostic codes available, we are limited in our ability to investigate the potential differences between bacterial, viral, and fungal LRTI etiologies. This has been added as a limitation to our study.

A rebuttal letter that responds to each point raised by the academic editor and reviewer(s). You should upload this letter as a separate file labeled ‘Response to Reviewers’.

A marked-up copy of your manuscript that highlights changes made to the original version. You should upload this as a separate file labeled ‘Revised Manuscript with Track Changes’.

An unmarked version of your revised paper without tracked changes. You should upload this as a separate file labeled ‘Manuscript’.

Journal Requirements:

Please ensure that your manuscript meets PLOS ONE’s style requirements, including those for file naming. The PLOS ONE style templates can be found at 

3. We note that you have referenced (Ho SL, Lu JQ, Buczek A, et al) which has currently not yet been accepted for publication. Please remove this from your References and amend this to state in the body of your manuscript: (Ho SL, Lu JQ, Buczek A, et al. [Submitted]”) as detailed online in our guide for authors

Thank you. It is replaced. 

4. We notice that your supplementary figures and tables are included in the manuscript file. Please remove them and upload them with the file type 'Supporting Information'. Please ensure that each Supporting Information file has a legend listed in the manuscript after the references list.

Thank you.

Reviewers' comments:

Reviewer's Responses to Questions

Comments to the Author

1. Is the manuscript technically sound, and do the data support the conclusions?

Reviewer #1: No

2. Has the statistical analysis been performed appropriately and rigorously?

Reviewer #1: No

3. Have the authors made all data underlying the findings in their manuscript fully available?

Reviewer #1: No

4. Is the manuscript presented in an intelligible fashion and written in standard English?

Reviewer #1: Yes

5. Review Comments to the Author

Reviewer #1: From the perspective of clinical real-world observation, this study conducted a detailed comparative analysis of population characteristics, basic diseases, laboratory examination indicators, a retrospective analysis of new liver diseases, and also an analysis of socio-economic factors, between COVID-19 and non-COVID-19 patients, as well as between COVID-19 patients and lower respiratory tract infected patients.

These analyses provide targeted data support for the prevention and management of newly diagnosed liver disease patients in the post pandemic era, and have very important practical significance.

It is almost a pleasant thing to read the text section of this article separately, except for the results section of abstract, which has a spelling error of "laower" instead of "lower".

However, the numerical values described in the results section of this article do not match the values in the chart or the abstract section, to the extent that they cannot be described in language. Listing them would be a huge workload and meaningless.

The numerical values of the main text, charts, and abstract, like three different studies, overshadow the topic which has great practical significance originally.

We thank the reviewer for their comments and apologize for the errors. We have updated the aforementioned sections.

Considering the large number of patients enrolled in this study and the long observation time, which may lead to a large number of institutions and personnel involved in this study, there may be difficulties in unified coordination in reality. Therefore, the authenticity of the data will not be discussed temporarily.

This is a retrospective study from a single health system which consisted of many hospitals and outpatient clinics. EMR data were extracted via OMOP data structure as reported previously (citations are provided). While it is not possible to check data manually for individual patients because of large sample size, we have carefully validated data as described in earlier studies. We added a statement to reflect your comment. 

Researchers must select a representative, read and review through the text, charts, and abstract, and confirm the consistency of the numerical descriptions of the three.

We apologize for the inconsistency. We have carefully reviewed to ensure consistency. 

1. This article lacks the table or graph of multiple regression analysis of COVID-19 and LRTI inpatients, which is similar to Table 2 showing the regression analysis of COVID-19 and non-COVID-19 patients.

We thank the reviewer for their comment. This has been added as Table 4. 

2. How can the median diagnostic time determined to be 6 months in the presentation of Supplementary Figure 1?

We thank the reviewer for their comment. We confirmed that the median was indeed 6 months (IQR=11 months). We also included mean and SD follow-up time.

---

## [Editor Report · Decision Letter 1]

22 Apr 2024

Incidence, characteristics, and risk factors of new liver disorders 3.5 years post COVID-19 pandemic in the Montefiore Health System in Bronx

PONE-D-23-43546R1

Dear Dr. Duong,

We’re pleased to inform you that your manuscript has been judged scientifically suitable for publication and will be formally accepted for publication once it meets all outstanding technical requirements.

Kind regards,

Benjamin M. Liu, MBBS, PhD, D(ABMM), MB(ASCP)

Academic Editor

PLOS ONE
---

## [Editor Report · Acceptance letter]

22 May 2024

PONE-D-23-43546R1 

PLOS ONE

Dear Dr. Duong, 

I'm pleased to inform you that your manuscript has been deemed suitable for publication in PLOS ONE. Congratulations! Your manuscript is now being handed over to our production team.

Kind regards, 

on behalf of

Dr. Benjamin M. Liu 

Academic Editor

PLOS ONE